# Pre-existing traits associated with Covid-19 illness severity

**Joseph E. Ebinger**[1,2☯], **Natalie Achamallah**[3,4☯], **Hongwei Ji**[5,6☯], **Brian L. Claggett**[6], **Nancy Sun**[1,2], **Patrick Botting**[1,2], **Trevor-Trung Nguyen**[1,2], **Eric Luong**[1,2], **Elizabeth H. Kim**[1,2], **Eunice Park**[7], **Yunxian Liu**[1,2], **Ryan Rosenberry**[1,2], **Yuri Matusov**[3,4], **Steven Zhao**[3,4], **Isabel Pedraza**[3,4], **Tanzira Zaman**[3,4], **Michael Thompson**[7], **Koen Raedschelders**[1,8], **Anders H. Berg**[9], **Jonathan D. Grein**[3,10], **Paul W. Noble**[3,11], **Sumeet S. Chugh**[1,2], **C. Noel Bairey Merz**[1,2,12], **Eduardo Marbán**[2], **Jennifer E. Van Eyk**[1,8,12], **Scott D. Solomon**[6], **Christine M. Albert**[1,2], **Peter Chen**[3,4,11☯] *, **Susan Cheng**[1,2,12☯] *

1 Department of Cardiology, Cedars-Sinai Medical Center, Los Angeles, California, United States of America, 2 Smidt Heart Institute, Cedars-Sinai Medical Center, Los Angeles, California, United States of America, 3 Department of Medicine, Cedars-Sinai Medical Center, Los Angeles, California, United States of America, 4 Division of Pulmonary and Critical Care Medicine, Cedars-Sinai Medical Center, Los Angeles, California, United States of America, 5 Shanghai Tenth People's Hospital, Tongji University, Shanghai, China, 6 Cardiovascular Division, Brigham and Women's Hospital, Boston, Massachusetts, United States of America, 7 Enterprise Information Systems Data Intelligence Team, Cedars-Sinai Medical Center, Los Angeles, California, United States of America, 8 Advanced Clinical Biosystems Institute, Cedars-Sinai Medical Center, Los Angeles, California, United States of America, 9 Department of Pathology and Laboratory Medicine, Cedars-Sinai Medical Center, Los Angeles, California, United States of America, 10 Department of Epidemiology, Cedars-Sinai Medical Center, Los Angeles, California, United States of America, 11 Women's Guild Lung Institute, Cedars-Sinai Medical Center, Los Angeles, California, United States of America, 12 Barbra Streisand Women's Heart Center, Cedars-Sinai Medical Center, Los Angeles, California, United States of America

☯ These authors contributed equally to this work.
* peter.chen@cshs.org (PC); susan.cheng@cshs.org (SC)

**Data Availability Statement:** The data that support the findings of this study are available from Cedars-Sinai Medical Center, upon reasonable request. The data are not publicly available due to the contents including information that could compromise

## Abstract

### Importance

Certain individuals, when infected by SARS-CoV-2, tend to develop the more severe forms of Covid-19 illness for reasons that remain unclear.

### Objective

To determine the demographic and clinical characteristics associated with increased severity of Covid-19 infection.

### Design

Retrospective observational study. We curated data from the electronic health record, and used multivariable logistic regression to examine the association of pre-existing traits with a Covid-19 illness severity defined by level of required care: need for hospital admission, need for intensive care, and need for intubation.

### Setting

A large, multihospital healthcare system in Southern California.

research participant privacy/consent. Please direct inquiries to: biodatacore@cshs.org.

**Funding:** This work was supported in part by the Erika J. Glazer Family Foundation (JEE; JEVE; CNBM; SC). The funder had no role in study design, data collection and analysis, decision to publish, or preparation of the manuscript. There was no additional external funding received for this study.

**Competing interests:** The authors have declared that no competing interests exist.

## Participants

All patients with confirmed Covid-19 infection (N = 442).

## Results

Of all patients studied, 48% required hospitalization, 17% required intensive care, and 12% required intubation. In multivariable-adjusted analyses, patients requiring a higher levels of care were more likely to be older (OR 1.5 per 10 years, P<0.001), male (OR 2.0, P = 0.001), African American (OR 2.1, P = 0.011), obese (OR 2.0, P = 0.021), with diabetes mellitus (OR 1.8, P = 0.037), and with a higher comorbidity index (OR 1.8 per SD, P<0.001). Several clinical associations were more pronounced in younger compared to older patients ($P_{interaction}$<0.05). Of all hospitalized patients, males required higher levels of care (OR 2.5, P = 0.003) irrespective of age, race, or morbidity profile.

## Conclusions and relevance

In our healthcare system, greater Covid-19 illness severity is seen in patients who are older, male, African American, obese, with diabetes, and with greater overall comorbidity burden. Certain comorbidities paradoxically augment risk to a greater extent in younger patients. In hospitalized patients, male sex is the main determinant of needing more intensive care. Further investigation is needed to understand the mechanisms underlying these findings.

## Introduction

The severe acute respiratory syndrome coronavirus-2 (SARS-CoV-2) is now well recognized as the cause of the coronavirus disease 2019 (Covid-19) global pandemic [1–3]. The rate of rise in Covid-19 infection and its associated outcomes in the United States is now comparable to rates observed in other severely affected countries such as China, Italy, and Spain [4–10]. The spread of Covid-19 in the United States has been especially pronounced in the states of California, New York, Michigan, Louisiana, and Washington [11]. Consistently reported across all regions is the observation that, of all individuals who become infected with SARS-CoV-2, a majority tend to have mild or no symptoms; however, an important minority will develop predominantly respiratory disease that can lead to critical illness and death [12–15]. Multiple, reports suggest that certain demographic and clinical characteristics may predispose infected persons to more severe manifestations of Covid-19, such as older age, male sex, and pre-existing hypertension, pulmonary disease, or cardiovascular disease [4, 16–19]. Given that these traits tend to cluster among the same persons, the relative contribution of each trait to the risk for developing more severe presentations of Covid-19 illness remains unclear.

We conducted a comprehensive investigation of the pre-existing demographic and clinical correlates of Covid-19 illness severity observed among patients evaluated for Covid-19 within our multi-site healthcare system in Los Angeles, California. We deliberately focused our study on pre-existing characteristics for two main reasons: first, we recognize that patients with Covid-19 illness can present early or late in the disease course, causing many clinical features to vary at the time of initial clinical encounter; and, second, we anticipate that ongoing public health efforts can be informed and augmented by understanding which predisposing factors

may render certain segments of the population at higher risk for the most morbid sequelae of SARS-CoV-2 infection.

## Methods

### Study sample

The Cedars-Sinai Health System is located in Los Angeles, California with a diverse catchment area of 1.8 million individuals, 33% of whom are over the age of 45 years and 80% identify as a racial or ethnic minority. The Cedars-Sinai Health System includes Cedars-Sinai Medical Center (CSMC), Marina Del Rey Hospital (MDRH), and affiliated clinics. For the current study, we included all patients who were found to have a laboratory confirmed diagnosis of SARS-CoV-2 infection while being evaluated or treated for signs or symptoms concerning for Covid-19 at CSMC or MDRH, beginning after the first confirmed case of community transmission was reported in the U.S. on February 26, 2020. Subsequently, the first laboratory confirmed Covid-19 case in our health system was on March 8, 2020. All laboratory testing for SARS-CoV-2 has been performed using reverse transcriptase polymerase chain reaction of extracted RNA from nasopharyngeal swabs. All patient testing was performed by the Los Angeles Department of Public Health until March 21, 2020, at which time the CSMC Department of Pathology and Laboratory Medicine began using the A*STAR FORTITUDE KIT 2.0 COVID-19 Real-Time RT-PCR Test (Accelerate Technologies Pte Ltd, Singapore). For the minority of patients in our study who had SARS-CoV-2 testing performed at an outside facility (3.6%), documentation of a positive test was carefully reviewed by our medical staff and considered comparable for accuracy.

### Data collection

For all patients considered to have Covid-19, based on direct or documented laboratory test result and suggestive signs and/or symptoms, we obtained information from the electronic health record (EHR) and verified data for the following demographic and clinical characteristics: age at the time of diagnosis; sex; race; ethnicity; smoking status defined as current versus prior, never, or unknown; comorbidities, including obesity, as clinically assessed and documented by a provider with ICD-10 coding; and, chronic use of angiotensin converting enzyme (ACE) inhibitor or angiotensin II receptor blocker (ARB) medications. Chronic use of ACE or ARB medications was verified by confirming presence of documented ongoing medication use in an outpatient provider's clinic note along with presence of an active outpatient prescription for the medication, both dated from prior to Covid-19 testing. We conducted iterative quality control and quality assurance analyses on all information extracted from the EHR; all data variables included in the main analyses were verified for completeness and accuracy through manual chart review, to avoid variable missingness or potential impact of inappropriate outliers in statistical modeling. Because presenting clinical measures such as vital signs and laboratory values can be highly variable, based on timing of the original clinical presentation, we elected to focus on pre-existing traits that may predispose to Covid-19 illness severity in a manner less dependent on the timing of patients presenting to medical care. To capture variation in relative comorbid status, in a way that is not captured by distinct medical history variables alone, we calculated the Elixhauser Comorbidity Index (ECI) with van Walraven weighting for all patients based on all available clinical data [20–23]. The ECI uses 31 categories to quantify a patient's burden of comorbid conditions and has been shown to outperform other indices in predicting adverse outcomes (S1 Table) [22–28]. For patients admitted to the hospital, length of stay, admission to an intensive care unit (ICU) and death were ascertained from time stamps recorded for admission, unit transfers, and discharge. Interventions such as

intubation and prone positioning were identified through time stamped orders in the EHR and verified by manual chart review. Dates and times of onset for reported or observed relevant signs and/or symptoms were also determined via manual chart review. All care was provided at the discretion of the treating physicians. Our outcomes for this study included: *severe illness* (defined as requiring any kind of hospital admission), *critical illness* (defined as the need for intensive care during hospitalization), and *respiratory failure* (defined as the need for intubation and mechanical ventilation). The CSMC institutional review board approved all protocols for the current study and waived the requirement for informed consent.

## Statistical analyses

For the total sample of Covid-19 patients, we used parametric tests to compare normally distributed continuous variables and non-normally distributed or categorical variables, respectively. We also used histograms to display age and sex distribution for the total cohort, the patients admitted but not requiring intensive care, and patients requiring intensive care at any time during hospitalization, and the patient requiring intubation and mechanical ventilation at any time during hospitalization. We used ordinal logistic regression to examine the associations between pre-existing characteristics (based on clinically relevant, non-missing data) and a primary outcome measure of illness severity, defined as an illness severity score. We constructed the illness severity score, with higher values assigned to needing more intensive levels of clinical care, based on the following stepwise categories: 0 = clinically deemed to not require admission; 1 = required hospital admission but never required intensive care; 2 = required intensive level care but never intubation; and, 3 = required intubation during hospitalization. We constructed age- and sex-adjusted models, from which significantly associated covariates (based on P<0.20) were selected for inclusion in the final multivariable-adjusted models, where appropriate (i.e. smaller sample sizes). Race was treated as a binary covariate: African American and non-African American. This approach was selected given the recently reported concerns of excess risk for African Americans [29], along with limited understanding of whether or not comorbidities contribute to this risk, in addition to the sample size for other race groups being too small for certain comparisons. Because hypertension and diabetes are not calculated as substantial contributors to the Elixhauser comorbidity index, we included each of these traits as separate additional covariates in all multivariable-adjusted analyses. We calculated the variance inflation factor (VIF) for each of the predictor variables to confirm absence of any substantial multicollinearity. In secondary analyses, we analyzed the associations of pre-existing patient characteristics with the distinct outcomes of needing any hospital admission (severe illness) and, in the cohort of all hospitalized patients representing an especially vulnerable population, the need for intensive care (critical illness) or intubation (respiratory failure). All analyses were performed using R, version 3.5.1 (R Foundation for Statistical Computing) and Stata, version 15 (StataCorp). For all final models, P values were 2-sided and considered significant at threshold level of 0.05.

## Results

Regional analyses showed that patients presented to our healthcare system from across a broad geographic catchment area in Los Angeles County (S1 Fig). The demographic and clinical characteristics of all patients in our study sample are shown in Table 1. Of all patients with pharmacologically treated hypertension, a minority were taking ACE inhibitor or ARB class agents and a majority were taking anti-hypertensive medications from alternate classes. Overall, almost half of patients (N = 214, 48%) were clinically assessed to require hospital admission, of whom over a third (N = 77; 36%) required intensive care and almost a quarter (24.3%)

**Table 1. Demographic and clinical characteristics of all patients with Covid-19.**

| | Total | Covid-19 Illness Severity | | | | P value* |
|---|---|---|---|---|---|---|
| | | *Not Admitted* | *Admitted, Non-ICU* | *ICU, Non-intubated* | *ICU, intubated* | |
| N | 442 | 228 | 137 | 25 | 52 | |
| Age, years, mean ± sd | 52.72 (19.65) | 43.16 (15.49) | 61.66 (19.54) | 64.24 (18.81) | 65.56 (15.16) | <0.001 |
| Male sex, n (%) | 256 (57.9) | 121 (53.1) | 78 (56.9) | 16 (64.0) | 41 (78.8) | 0.007 |
| Smoker, n (%) | 16 (5.5) | 13 (8.5) | 3 (3.2) | 0 (0.0) | 0 (0.0) | 0.104 |
| Ethnicity, n (%) | | | | | | 0.012 |
| Non-Hispanic | 341 (77.1) | 167 (73.2) | 112 (81.8) | 21 (84.0) | 41 (78.8) | |
| Hispanic | 68 (15.4) | 33 (14.5) | 22 (16.1) | 4 (16.0) | 9 (17.3) | |
| Race, n (%) | | | | | | <0.001 |
| White | 283 (64.0) | 136 (59.6) | 99 (72.3) | 17 (68.0) | 31 (59.6) | |
| African American | 58 (13.1) | 19 (8.3) | 21 (15.3) | 4 (16.0) | 14 (26.9) | |
| Asian | 35 (7.9) | 25 (11.0) | 7 (5.1) | 3 (12.0) | 0 (0.0) | |
| Other | 37 (8.4) | 23 (10.1) | 8 (5.8) | 1 (4.0) | 5 (9.6) | |
| Obesity, n (%) | 71 (16.1) | 27 (11.8) | 27 (19.7) | 4 (16.0) | 13 (25.0) | 0.059 |
| Hypertension, n (%) | 161 (36.4) | 44 (19.3) | 68 (49.6) | 20 (80.0) | 29 (55.8) | <0.001 |
| Diabetes mellitus, n (%) | 84 (19.0) | 18 (7.9) | 40 (29.2) | 10 (40.0) | 16 (30.8) | <0.001 |
| Elixhauser comorbidity score, mean ± sd | 6.32 (10.78) | 1.31 (4.35) | 10.62 (12.17) | 17.52 (16.54) | 11.62 (12.05) | <0.001 |
| Prior myocardial infarction or heart failure, n (%) | 49 (11.1) | 4 (1.8) | 27 (19.7) | 8 (32.0) | 10 (19.2) | <0.001 |
| Prior COPD or asthma, n (%) | 70 (15.8) | 27 (11.8) | 28 (20.4) | 6 (24.0) | 9 (17.3) | 0.101 |
| ACE inhibitor use, n (%) | 31 (7.0) | 11 (4.8) | 15 (10.9) | 1 (4.0) | 4 (7.7) | 0.15 |
| Angiotensin receptor blocker use, n (%) | 41 (9.3) | 13 (5.7) | 15 (10.9) | 4 (16.0) | 9 (17.3) | 0.026 |

* P values are for between-group comparisons using the ANOVA test for continuous variables and the chi-square test for categorical variables.

A total of 29 patients were missing race data and 33 patients were missing ethnicity data.

required intubation. In unadjusted analyses, the patients who were more likely to require higher levels of care tended to be older, male, African American, and with known hypertension, diabetes mellitus, higher Elixhauser comorbidity index, and have prior myocardial infarction or heart failure (Table 1). The number of men with confirmed Covid-19 infection outnumbered women in nearly all age groups; this sex difference was more pronounced among patients requiring hospitalization and particularly among patients requiring intensive care or intubation (Fig 1). We also observed a consistently higher rate of greater illness severity among African Americans compared to persons of other racial groups (Fig 2).

For the primary outcome of illness severity, categorized by escalating levels of care (i.e., hospitalization, intensive care, intubation), the pre-existing characteristics that demonstrated statistical significance in age- and sex-adjusted models included older age, male sex, African American race, obesity, hypertension, diabetes mellitus, and the Elixhauser comorbidity score (Table 2; Fig 3). The associations that remained significant in the fully-adjusted multivariable model included older age (odds ratio [OR] 1.49 per 10 years, 95% confidence interval [CI] 1.30–1.70, P<0.001), male sex (OR 2.01, 95% CI 1.34–3.04, P = 0.001), African American race (OR 2.13, 95% CI 1.19–3.83, P = 0.011), obesity (OR 1.95, 95% CI 1.11–3.42, P = 0.021), diabetes mellitus (OR 1.77, 95% CI 1.03–3.03, P = 0.037) and the comorbidity score (OR 1.77 per SD, 95% CI 1.37–2.28, P<0.001). We also observed a trend towards lower severity of illness among patients chronically treated with ACE inhibitor therapy, with OR 0.48 (95% CI 0.22–1.04; P = 0.06). Each estimated OR value represents the increment in higher (or lower) odds of a patient requiring a next higher level of care, for every unit difference in a continuous variable

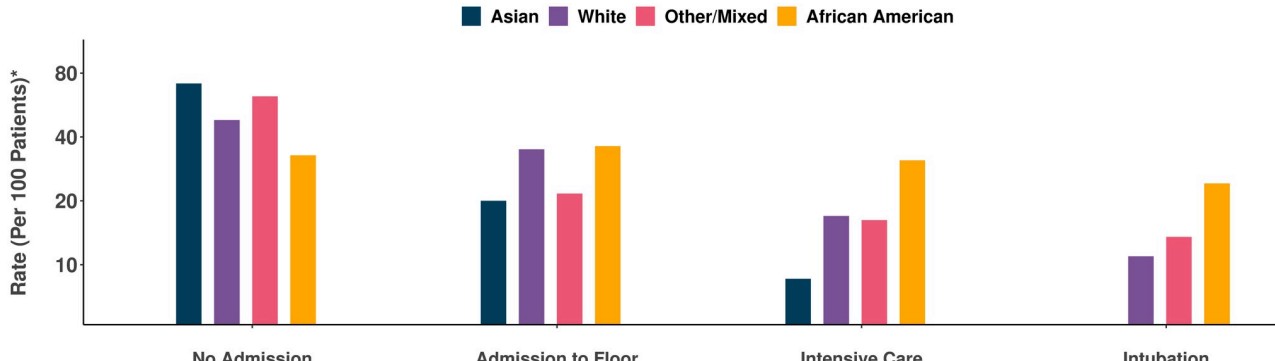

**Fig 1. Age and sex distribution of patients with Covid-19, stratified by admission status.** The frequency of laboratory confirmed Covid-19 was higher in males compared to females particularly among individuals requiring hospital admission, individuals with critical illness (requiring intensive care), and individuals with respiratory failure (requiring intubation).

**Fig 2. Rates of clinical outcomes of all patients with Covid-19, stratified by race.** The frequency of African Americans manifesting more severe forms of Covid-19 illness, requiring higher levels of clinical care, was greater than that for other racial groups. *Rate was calculated as proportion of cases within each racial group.

**Table 2. Characteristics associated with overall Covid-19 illness severity* in the total sample (N = 442).**

|  | Age- and Sex-Adjusted Models | | Multivariable-Adjusted Model† | |
|---|---|---|---|---|
|  | OR (95% CI) | *P value* | OR (95% CI) | *P value* |
| **Age, per 10 years** | 1.68 (1.52,1.87) | <0.001 | **1.49 (1.30,1.70)** | **<0.001** |
| **Male sex** | 1.87 (1.26,2.77) | 0.002 | **2.01 (1.34,3.04)** | **0.001** |
| **African American race∫** | 2.46 (1.45,4.18) | <0.001 | **2.13 (1.19,3.83)** | **0.011** |
| Hispanic ethnicity | 1.54 (0.91,2.60) | 0.11 | 1.39 (0.79,2.45) | 0.26 |
| **Obesity** | 1.96 (1.19,3.24) | 0.009 | **1.95 (1.11,3.42)** | **0.021** |
| Hypertension | 1.97 (1.27,3.05) | 0.003 | 1.19 (0.71,1.99) | 0.52 |
| **Diabetes mellitus** | 2.25 (1.41,3.57) | 0.001 | **1.77 (1.03,3.03)** | **0.037** |
| **Elixhauser comorbidity score, per SD** | 1.63 (1.33,2.01) | <0.001 | **1.77 (1.37,2.28)** | **<0.001** |
| Prior myocardial infarction or heart failure | 1.72 (0.96,3.09) | 0.07 | 0.56 (0.27,1.18) | 0.13 |
| Prior COPD or asthma | 1.23 (0.75,2.03) | 0.41 | 0.76 (0.44,1.31) | 0.34 |
| ACE inhibitor use | 0.69 (0.35,1.38) | 0.29 | 0.48 (0.22,1.04) | 0.06 |
| Angiotensin receptor blocker use | 1.18 (0.63,2.19) | 0.61 | 1.05 (0.54,2.06) | 0.89 |

*The primary outcome of Covid-19 illness severity score in the total sample was defined as an ordinal variable wherein: 0 = referent, 1 = required admission but never ICU level care, 2 = required ICU level care but never intubated, 3 = required intubation.

† All listed covariates shown were included in the full multivariable-adjusted model.

∫The referent is non-African American race.

(e.g. per 10 years of age) or for presence versus absence of a given categorical variable (e.g. male sex). In effect, every 10 years of older age was associated with ~1.5-fold higher odds of requiring a higher level of care, and being male versus female was associated with a ~2-fold higher odds of requiring higher level care. We used the Brant method to test the proportional odds assumption for consistency of associations across our ordinal outcome; these analyses revealed no substantial qualitative violations, but did indicate that the Elixhauser score was predominantly associated with the specific outcomes of admission versus non-admission (OR 4.34, P<0.001) and need for intensive care versus no intensive care need (OR 1.55, P = 0.008) that with the less frequent outcome of needing intubation versus no need for intubation (OR 1.24, P = 0.25).

For the specific outcome of needing any hospital admission, the pre-admission characteristics that demonstrated statistical significance included older age, male sex, African American race, obesity, hypertension, diabetes mellitus, the Elixhauser comorbidity index, and prior myocardial infarction or heart failure (S2 Table). In the multivariable model adjusting for all key covariates, the pre-existing traits that remained significantly associated with needing any hospital admission were older age, diabetes mellitus, and higher comorbidity index.

Among the patients whose illness severity required hospitalization, male sex was associated with the outcome of requiring further escalating levels of care (i.e., intensive care and intubation) (Table 3; Fig 3). In the multivariable model adjusting for key covariates, male sex remained the single most important risk marker of requiring higher-level care (OR 2.53, 95% CI 1.36–4.70, P = 0.003). The results for male sex were similar for the individual outcomes of requiring intensive care or intubation (S3 Table). We again observed a trend towards lower need for admission to the intensive care unit among patients chronically taking an ACE inhibitor (OR 0.38, 95% CI 0.13–1.17, P = 0.09), and greater need for intubation among African Americans patients (OR 2.14, 95% CI 0.99–4.64, P = 0.053).

In secondary analyses, we used multiplicative interaction terms to assess for effect modification for associations observed in the main analyses (S4 Table). While considered exploratory

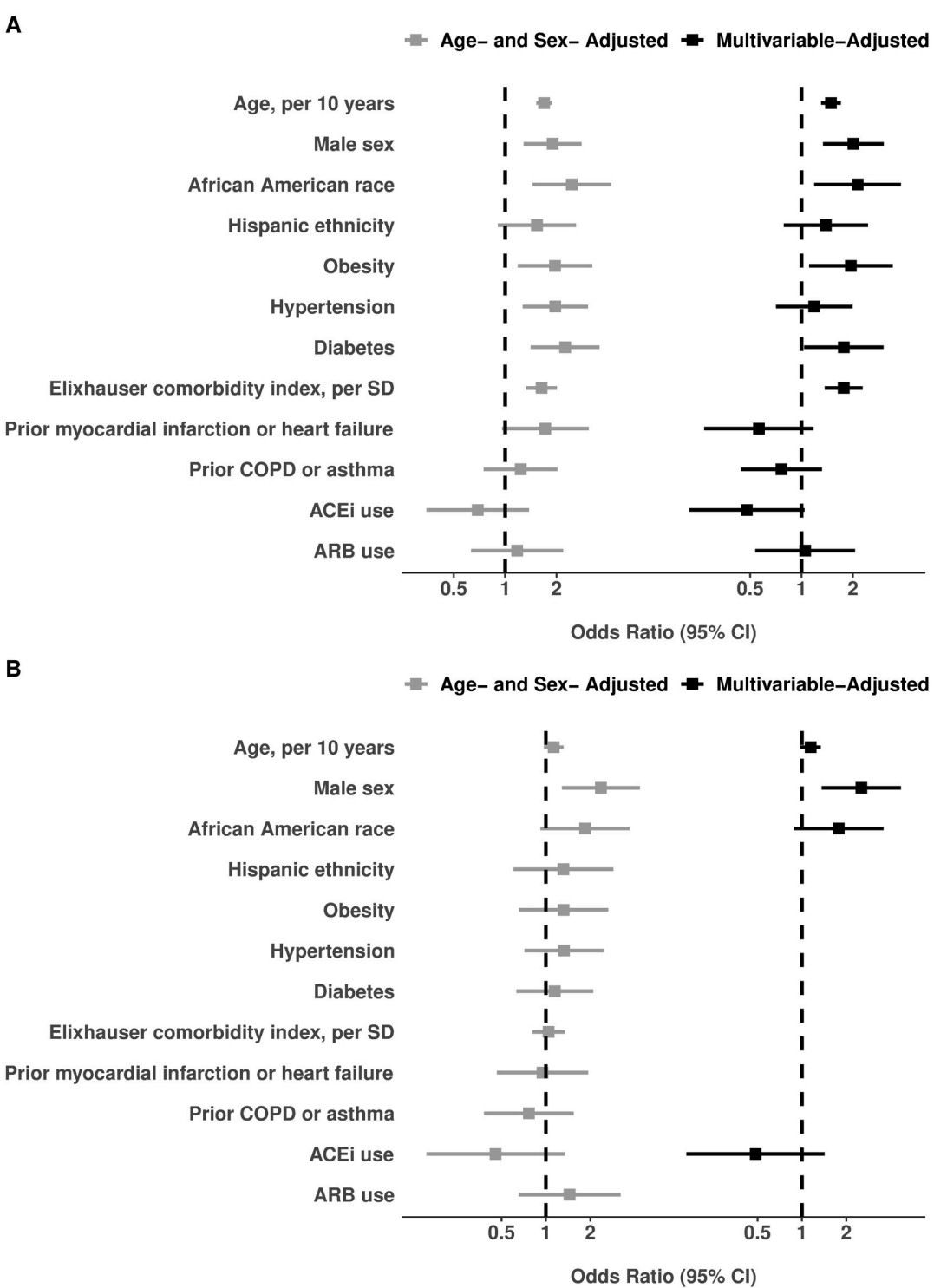

**Fig 3. Characteristics associated with overall Covid-19 illness severity.** Results for the total sample of N = 442 admitted and non-admitted patients are shown in **Panel A** (all listed covariates shown were in the full multivariable-adjusted model). Results for the N = 214 admitted patients are shown in **Panel B** (to avoid model overfitting given the smaller sample size, covariates included in the multivariable model were selected from age- and sex-adjusted models based on significance with P<0.20).

**Table 3. Characteristics associated with Covid-19 illness severity among all hospitalized patients.**

|  | Age- and Sex-Adjusted Models | | Multivariable-Adjusted Model† | |
|---|---|---|---|---|
|  | OR (95% CI) | *P value* | OR (95% CI) | *P value* |
| Age, per 10 years | 1.13 (0.97,1.32) | 0.12 | 1.14 (0.98,1.34) | 0.09 |
| **Male sex** | 2.36 (1.29,4.34) | 0.006 | **2.53 (1.36,4.70)** | **0.003** |
| African American race | 1.85 (0.92,3.71) | 0.08 | 1.78 (0.88,3.58) | 0.11 |
| Hispanic ethnicity | 1.31 (0.60,2.86) | 0.49 | - | - |
| Obesity | 1.32 (0.66,2.65) | 0.44 | - | - |
| Hypertension | 1.33 (0.72,2.46) | 0.37 | - | - |
| Diabetes mellitus | 1.15 (0.63,2.09) | 0.65 | - | - |
| Elixhauser comorbidity score, per SD | 1.04 (0.81,1.34) | 0.74 | - | - |
| Prior myocardial infarction or heart failure | 0.95 (0.47,1.94) | 0.89 | - | - |
| Prior COPD or asthma | 0.77 (0.38,1.54) | 0.46 | - | - |
| ACE inhibitor use | 0.45 (0.15,1.34) | 0.15 | 0.48 (0.16,1.43) | 0.19 |
| Angiotensin receptor blocker use | 1.45 (0.65,3.21) | 0.36 | - | - |

*The secondary outcome of Covid-19 illness severity score in hospitalized patients was defined as an ordinal variable wherein: 1 = referents required admission but never ICU level care, 2 = required ICU level care but never intubate, 3 = required intubation.

†To avoid model overfitting given the sample size, covariates included in the multivariable model were selected from age- and sex-adjusted models based on significance with $P<0.20$.

or hypothesis generating analyses, we found several interactions of potential interest (Fig 4). In particular, the associations of Hispanic ethnicity, obesity, diabetes, and Elixhauser comorbidity index with the primary outcome appeared paradoxically more pronounced in younger compared to older individuals (S5 Table). By contrast, the primary outcome was more pronounced among older compared to younger African Americans. Also paradoxically, hypertension appeared associated with greater risk in non-obese patient and with lower risk in obese patients. We repeated all main analyses with additional adjustment for smoking status in the subset of patients with available data on smoking; in these models, all significant results remained unchanged (S6 and S7 Tables).

## Discussion

We examined the pre-existing characteristics associated with severity of Covid-19 illness, as observed thus far in our healthcare system located in Los Angeles, California. We found that almost half of patients presenting for evaluation and then confirmed to have Covid-19 were clinically assessed to require hospital admission. These higher risk individuals were more likely to be older, male, African American, obese, and have diabetes mellitus in addition to a greater overall burden of medical comorbidities. Notably, chronic use of an ACE inhibitor appeared related to lower illness severity, in the absence of a similar finding for ARB use. Among all individuals requiring inpatient care for Covid-19, male patients had a greater than 2.5-fold odds of needing intensive care and a 3.0-fold odds of needing intubation. All of our findings were observed even after accounting for co-existing risk factors and chronic medical conditions.

Recognizing that patients with Covid-19 illness can present with clinical features that vary based on timing of the index encounter, we sought to identify the pre-existing traits that render some individuals at highest risk for developing the more severe forms of Covid-19 illness once contracted. In our U.S. based metropolitan community, we observed that both obesity

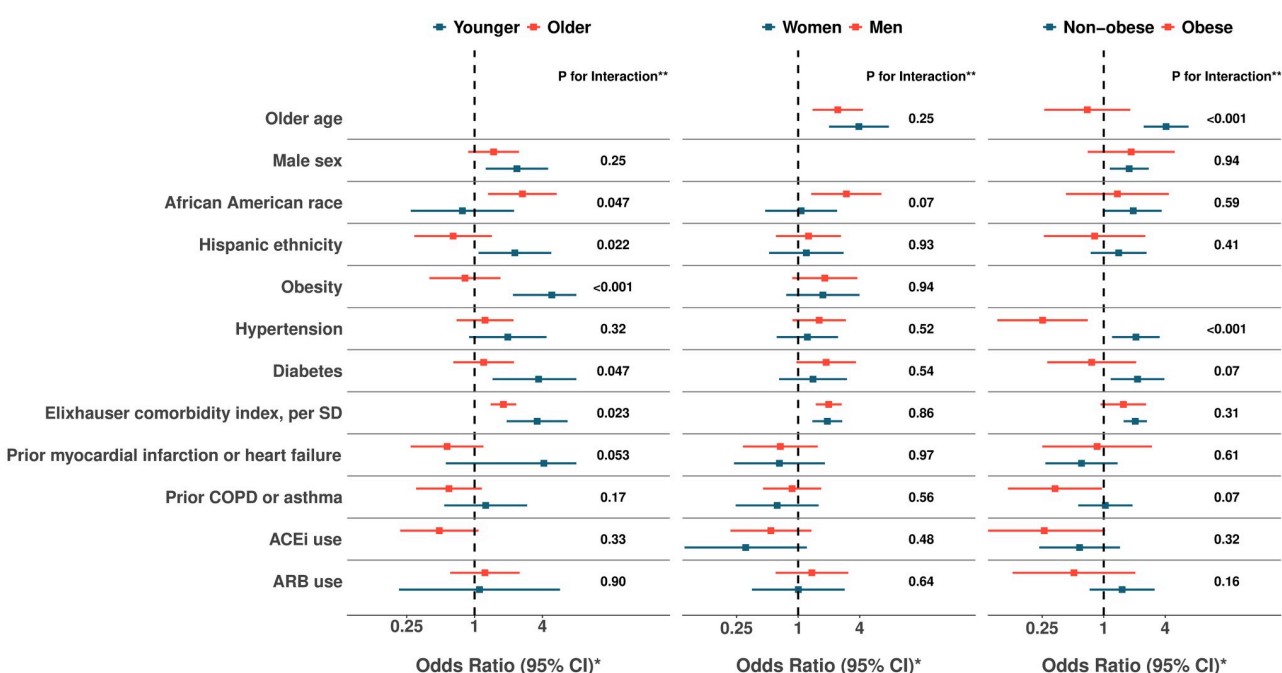

**Fig 4. Associations with overall Covid-19 illness severity, stratified by subgroups.** Relative risks associated with illness severity score are shown for all associations observed in the total sample (N = 442), stratified by subgroups defined by age (younger vs. older than median age 52 years), sex, and obesity (BMI ≥30 kg/m²). *The primary outcome of Covid-19 illness severity score in the total sample was defined as an ordinal variable wherein: 0 = referent, 1 = required admission but never ICU level care, 2 = required ICU level care but never intubated, 3 = required intubation. **P for interaction values were calculated from likelihood ratio test between models with and without the interaction term. For each variable in the list, age (<versus ≥median age of 52 years), sex, and obesity interaction terms are implemented in multivariable adjusted models, with other covariates representative of the entire cohort.

and diabetes mellitus are predisposing factors associated with a greater odds of needing hospital admission for Covid-19 but not of requiring further escalation of care; this finding is consistent with emerging reports of obesity and diabetes mellitus each being associated with a greater risk for pneumonia due to Covid-19 as well as other community-acquired viral agents—particularly in areas of the world where obesity is prevalent [30–35]. Also consistent with worldwide reports, we observed that older age is a significant predisposing risk factor for greater Covid-19 illness severity in multivariable-adjusted models; this finding may represent an age-related immune susceptibility that is not completely captured by even a comprehensive comorbidity measure such as the Elixhauser index. Notwithstanding an overall age association in the expected direction of risk, we also found a paradoxical age interaction for certain key correlates. In effect, presence of obesity, diabetes, or an elevated overall comorbidity index were each associated with greater Covid-19 illness severity in younger (i.e. <52 years) compared to older age groups. While unexpected, this finding is actually consistent with the known reduction of ACE2 expression with advancing age, a phenomenon that has been proposed as a major contributor to the broad susceptibility to Covid-19 seen in younger to middle aged individuals across the population at large [36].

Consistent with worldwide reports, we found that the association of male sex with greater odds for every metric of Covid-19 illness severity was especially prominent—and this was not explained by age variation, risk factors, or comorbidities [37]. Reasons for the male predominance of illness severity remain unclear. Although ACE2 genetic expression is on the *X* chromosome, evidence to date would suggest relatively comparable expression levels between sexes

[38, 39], albeit with some potential for variation in relation to differences in sex hormones; select animal studies have shown increased ACE2 activity in the setting of ovariectomy and the opposite effect with orchietomy [40, 41]. While there remains scant data currently available to explain sex differences for Covid-19, male sex bias was also observed for SARS and MERS [42, 43]. Similar to the findings in our study, this increase risk was not attributable to a greater prevalence of smoking among men. Notably, prior murine studies have also demonstrated male versus female bias in susceptibility to SARS-CoV infection, which may be related to the effects of sex-specific steroids and *X*-linked gene activity on modulation of both the innate and adaptive immune response to viral infection [44]. Further research specific to the sexual dimorphism seen in SARS-CoV-2 susceptibility is needed.

In our U.S. based metropolitan community, we also observed racial and ethnic patterns of susceptibility to greater Covid-19 illness severity. Specifically, we found that African Americans were at greater risk for needing higher levels of care overall, and this vulnerability appeared more pronounced in older age and among men. Although an overall risk association was not seen for Hispanic ethnicity, there was a trend towards greater Covid-19 illness severity in younger aged compared to older aged Hispanic/Latino persons. A recent national report from the CDC also suggests overall higher rates of Covid-19 susceptibility in African Americans, and our findings confirm this trend exists even after adjusting for age, risk factors, and comorbidities [45]. In addition to the effects of unmeasured socioeconiomic and healthcare access variables, racial/ethnic disparities in Covid-19 illness severity may relate to yet unidentified host-viral susceptibility factors that could also be contributing to heterogeneity of community transmission seen across regions worldwide and populations at large [29].

The use of ACE inhibitor or angiotensin receptor blocker (ARB) medications has been a focus of attention given that these agents may upregulate expression of ACE2, the viral point of entry into cells [46] and alveolar type 2 epithelial cells in particular [47]. Alternatively, these agents may confer benefit, given that SARS-CoV-2 appears to reduce ACE2 activity and lead to potentially unopposed excess renin-angiotensin-aldosterone activation [36, 46, 48]. Although we observed a non-significant trend in association of chronic ACE inhibitor treatment with lower Covid-19 illness severity, we found evidence of neither risk nor benefit with ARBs. Together, our findings are supportive of current recommendations to not discontinue chronic ACE inhibitor or ARB therapy for patients with appropriate indications for these medications.

Several limitations of our study merit consideration. Our cohort included all individuals who underwent laboratory testing for Covid-19 and not individuals who did not undergo testing; thus, our study results are derived from individuals presenting with symptoms that were deemed severe enough to warrant testing. All data including past medical history data were collected from the EHR and, thus, subject to coding bias and variations in reporting quality. To minimize the potential effects of these limitations that are inherent to EHR data, we performed iterative quality checks on the dataset and conducted manual chart review to verify values for key variables. We recognize that the illness severity outcomes defined as clinically ascertained need for hospital admission, ICU level care, and intubation, may vary from practice to practice. As in many other U.S. medical centers affected by the Covid-19 pandemic, our clinical staff have been practicing under institutional guidance to conserve resources and we anticipate that the thresholds for escalating care are likely comparable; thresholds for admission, transfer to intensive care, and intubation may be different in more resource constrained environments. Given the relatively small number of observed in-hospital deaths (N = 11), and thus limited statistical power to detect associations, we deferred analyses of pre-existing characteristics and mortality risk to future investigations. The modest size of this early analysis of our growing clinical cohort may have limited our ability to detect potential additional

predictors of Covid-19 illness severity, as well as potential interactions or effect modification relevant to the outcomes; thus, further investigations are needed in larger sized samples. Finally, our results are derived from a single healthcare system, albeit multi-center and serving a large catchment of the diverse population of Los Angeles, California. Additional studies are needed to examine the extent to which our findings are generalizable to other populations affected by Covid-19.

In summary, we found that among patients tested and managed for laboratory confirmed Covid-19 in our healthcare system to date, approximately half require admission for inpatient hospital care. These individuals are more likely to be older, male, African American, obese, and with known diabetes mellitus as well as a greater overall burden of medical comorbidities. Well over a third of hospitalized patients require intensive care, with a substantial proportion needing intubation and mechanical ventilation for respiratory failure. Among hospitalized patients, the highest risk individuals were more likely to be predominantly men of any age or race—for reasons not explained by comorbidities. Further investigations are needed to understand the mechanisms underlying these associations and, in turn, determine the most optimal approaches to attenuating adverse outcomes for all persons at risk.

## Supporting information

**S1 Table. Elixhauser comorbidity index and van Walraven weights.**
(DOCX)

**S2 Table. Characteristics associated with need for any hospitalization in all patients with Covid-19.**
(DOCX)

**S3 Table. Characteristics associated with distinct outcomes in patients hospitalized for Covid-19.**
(DOCX)

**S4 Table. Age, sex, and body mass index interactions with characteristics associated with overall Covid-19 illness severity in the total sample.**
(DOCX)

**S5 Table. Age, sex, and obesity stratified associations with overall Covid-19 illness severity in the total sample.**
(DOCX)

**S6 Table. Characteristics associated with overall Covid-19 illness severity\* in the total sample.**
(DOCX)

**S7 Table. Characteristics associated with Covid-19 illness severity among all hospitalized patients.**
(DOCX)

**S1 Fig. Los Angeles county regional distribution of all patients with Covid-19.** The patients treated in our healthcare system for Covid-19 illness presented from across a diverse regional distribution of residential locations across Los Angeles County. The map shown was generated using ArcGIS software by Esri.
(TIF)

## Acknowledgments

We are grateful to all the front-line healthcare workers in our healthcare system who continue to be dedicated to delivering the highest quality care for all patients.

## Author Contributions

**Conceptualization:** Jonathan D. Grein, Susan Cheng.

**Data curation:** Joseph E. Ebinger, Natalie Achamallah, Hongwei Ji.

**Formal analysis:** Joseph E. Ebinger, Hongwei Ji, Brian L. Claggett, Susan Cheng.

**Funding acquisition:** Susan Cheng.

**Investigation:** Joseph E. Ebinger, Hongwei Ji, Brian L. Claggett, Nancy Sun, Patrick Botting, Trevor-Trung Nguyen, Eric Luong, Elizabeth H. Kim, Eunice Park, Yunxian Liu, Ryan Rosenberry, Yuri Matusov, Steven Zhao, Isabel Pedraza, Tanzira Zaman, Michael Thompson, Koen Raedschelders, Anders H. Berg, Jonathan D. Grein, Paul W. Noble, Sumeet S. Chugh, C. Noel Bairey Merz, Eduardo Marbán, Jennifer E. Van Eyk, Scott D. Solomon, Christine M. Albert, Susan Cheng.

**Methodology:** Susan Cheng.

**Project administration:** Susan Cheng.

**Resources:** Susan Cheng.

**Supervision:** Peter Chen, Susan Cheng.

**Validation:** Joseph E. Ebinger, Hongwei Ji.

**Writing – original draft:** Joseph E. Ebinger, Hongwei Ji, Susan Cheng.

**Writing – review & editing:** Joseph E. Ebinger, Natalie Achamallah, Hongwei Ji, Brian L. Claggett, Nancy Sun, Patrick Botting, Trevor-Trung Nguyen, Eric Luong, Elizabeth H. Kim, Eunice Park, Yunxian Liu, Ryan Rosenberry, Yuri Matusov, Steven Zhao, Isabel Pedraza, Tanzira Zaman, Michael Thompson, Koen Raedschelders, Anders H. Berg, Jonathan D. Grein, Sumeet S. Chugh, C. Noel Bairey Merz, Eduardo Marbán, Jennifer E. Van Eyk, Scott D. Solomon, Christine M. Albert, Peter Chen, Susan Cheng.

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
