## [Decision Letter · Decision Letter 0]

15 May 2020

PONE-D-20-12746

Pre-Existing Traits Associated with Covid-19 Illness Severity

PLOS ONE

Dear Dr. Cheng,

Thank you for submitting your manuscript to PLOS ONE. After careful consideration, we feel that it has merit but does not fully meet PLOS ONE’s publication criteria as it currently stands. Therefore, we invite you to submit a revised version of the manuscript that addresses the points raised during the review process.

Both reviewers raised several concerns especially regarding collection, analysis, and presentation of data. The authors need to be effectively respond to their comments in the revision.

We would appreciate receiving your revised manuscript by Jun 28 2020 11:59PM. To enhance the reproducibility of your results, we recommend that if applicable you deposit your laboratory protocols in protocols.io, where a protocol can be assigned its own identifier (DOI) such that it can be cited independently in the future. For instructions see: http://journals.plos.org/plosone/s/submission-guidelines#loc-laboratory-protocols

We look forward to receiving your revised manuscript.

Kind regards,

Yu Ru Kou, PhD

Academic Editor

PLOS ONE

Journal Requirements:

2. In your ethics statement in the Methods section and in the online submission form, please provide additional information about the data used in your retrospective study. Specifically, please ensure that you have discussed whether all data were fully anonymized before you accessed them and/or whether the IRB or ethics committee waived the requirement for informed consent. If patients provided informed written consent to have data from their medical records used in research, please include this information.

4. Please amend the manuscript submission data (via Edit Submission) to include authors Joseph E. Ebinger, MD, MS, Natalie Achamallah, MD, MS, Hongwei Ji, MD, Brian L. Claggett, PhD, Nancy Sun, MPS, Patrick Botting, MSPH, Trevor-Trung Nguyen, BS, Eric Luong, MPH, Elizabeth H. Kim, BA, Eunice Park, BS, Yunxian Liu, MS, PhD, Ryan Rosenberry, PhD, Yuri Matusov, MD, Steven Zhao, MD, Isabel Pedraza, MD, Tanzira Zaman, MD, Michael Thompson, BS, MS, Koen Raedschelders, PhD, Anders H. Berg, MD, PhD, Jonathan D. Grein, MD, Paul W. Noble, MD, Sumeet S. Chugh, MD, C. Noel Bairey Merz, MD, Eduardo Marbán, MD, PhD, Jennifer E. Van Eyk, PhD, Scott D. Solomon, MD, Christine M. Albert, MD, MPH and Peter Chen, MD.

6. We note that Supplemental Figure 1 in your submission contain map image which may be copyrighted. All PLOS content is published under the Creative Commons Attribution License (CC BY 4.0), which means that the manuscript, images, and Supporting Information files will be freely available online, and any third party is permitted to access, download, copy, distribute, and use these materials in any way, even commercially, with proper attribution. For these reasons, we cannot publish previously copyrighted maps or satellite images created using proprietary data, such as Google software (Google Maps, Street View, and Earth). For more information, see our copyright guidelines: http://journals.plos.org/plosone/s/licenses-and-copyright.

6.1    You may seek permission from the original copyright holder of Supplemental Figure 1 to publish the content specifically under the CC BY 4.0 license.

6.2.   If you are unable to obtain permission from the original copyright holder to publish these figures under the CC BY 4.0 license or if the copyright holder’s requirements are incompatible with the CC BY 4.0 license, please either i) remove the figure or ii) supply a replacement figure that complies with the CC BY 4.0 license. Please check copyright information on all replacement figures and update the figure caption with source information. If applicable, please specify in the figure caption text when a figure is similar but not identical to the original image and is therefore for illustrative purposes only.

Reviewers' comments:

Reviewer's Responses to Questions

**Comments to the Author**

1. Is the manuscript technically sound, and do the data support the conclusions?

Reviewer #1: Yes

Reviewer #2: Partly

2. Has the statistical analysis been performed appropriately and rigorously? 

Reviewer #1: Yes

Reviewer #2: Yes

3. Have the authors made all data underlying the findings in their manuscript fully available?

Reviewer #1: Yes

Reviewer #2: Yes

4. Is the manuscript presented in an intelligible fashion and written in standard English?

Reviewer #1: Yes

Reviewer #2: Yes

5. Review Comments to the Author

Reviewer #1: The authors investigated the association between patients’ preexisting traits and the coronavirus disease (COVID-19) severity as defined by the level of required care, including the needs for hospital admission, intensive care, and intubation, in patients with laboratory-confirmed COVID-19. The study was conducted in the Cedars-Sinai Health System (Cedars-Sinai Medical Center, Marina Del Rey Hospital, and affiliated clinics) in Los Angeles, California. The present study concluded that the COVID-19 severity was greater in patients who were older, male, and African American, and had obesity, diabetes, and greater overall comorbidity burden. This study evaluated an important clinical issue and was well conducted, and the writing of the manuscript is coherent and well organized. I only have a few questions and comments as follows:

1. In the Data Collection section, the authors mentioned that they obtained data on length of stay, death, and vital signs/laboratory diagnostics assessed within 1 week prior to presentation. I did not find any analytical results related to these variables (as shown in Table 1, baseline characteristics, or adjustment in the regression models). How were these data utilized in the study?

2. As this study was conducted on the basis of electronic health records, I assumed that some missing data inevitably existed. I suggest briefly summarizing how much data were missing and how the authors treated the missing data in their analyses.

3. What model was chosen for performing the ordinal logistic regression analysis? If the proportional odds model was selected, did the authors check the proportional odds assumption?

4. Was there collinearity between any of the predictor variables (e.g., ethnicity and race, which were treated as two different variables) listed in Table 1? As all the variables listed in Table 1 were included in the multivariable regression models and each OR was calculated and interpreted for each independent variable in this study, the authors should consider checking the assumption of no multi-collinearity, which is one of the assumptions when performing ordinal logistic regression analyses.

5. In Table 2, why did you calculate only the OR of the African American race? There are other races such as Asian and others (I assume the white race as reference) shown in Table 1.

6. As the authors claimed a trend toward lower illness severity among the patients chronically treated with angiotensin-converting enzyme inhibitor (ACEI) therapy, with an OR of 0.48 (P = 0.06), I suggest describing how they defined “chronically treated with drugs (ACEI and ARB)” in the Methods section, as these findings are interesting and clinically relevant.

7. On page 10, first paragraph: “…ACE inhibitor (OR 0.38, 95% CI 0.13-0.17, P=0.09)…,” I believe there was an error in the confidence interval. Please correct it. Please check the accuracy of all the statistical values presented in this manuscript.

Reviewer #2: This study investigates the association between pre-existing diseases (comorbidities) and the severity of COVID19 (n=442). The authors defined the outcome as 0, 1, 2, 3 which is an ordinal scale reflecting the severity of COVID19. Based upon the multivariate ordinal logistic regression analysis, they found older age, male, race, obese, DM and high ECI were significantly associated with the severity of COVID 19. The topic is important for the COVID 19 pandemic, but I have some concerns about the methods:

Major comment:

1. Recent studies indicate that SARS-CoV-2 might enter host cells by binding angiotensin-converting enzyme 2 (ACE2). However, the theme of this study is to investigate the risk of severity in relation to pre-existing traits rather ACE2. In this sample, the age and race differences exist among the 4 groups. If ACEI or ARB prescriptions are related to age or race, this might cause the bias. Further, the current sample size of ACEI and ARB is too few to confirm ACE2 hypothesis. I recommend the authors should focus on the theme of this study. Particularly, the total percentage of ACEI (n=31) and ARB (n=41) is 16% which is much lower than hypertension 36%. Why is it?

2. Because there are different forms of ordinal logistic regression models to take care of ordinal outcome (0= not require admission; 1=required hospitaladmission without intensive care; 2 = required intensive level care without intubation; and, 3 = required intubation during hospitalization), the authors should clearly describe which model is used and examine whether the assumption (e.g., proportional odds) holds or not.

3. The primary analysis is ordinal logistic regression (the outcome is 0,1,2,3), and the secondary analysis is logistic regression analysis (the focus is for specific outcome, a binary variable). The authors need to explain the meaning of the estimated odds ratio from each analysis. Also, since ECI depends on 31 comorbidities including DM, Hypertension and obesity, do the authors assess the collinearity between the covariates?

4. The authors need to provide the details of interaction models. Because of small sample size (n=442, particularly n=52 for patients required intubation), the power to detect 11 interaction terms might be low (in Suppl. Tables 4-5). Please explain it better.

5. In Table1: according to the Covid-19 Illness Severity outcome ( i.e., 0, 1, 2, 3), the authors should compare the sample characteristics among the 4 disjoint groups to fit their goal. Smoking status could be added although the data might not be complete. Remove “unknown” category for ethnicity and race.

Minor comments:

6. Methods: Data collection

The details of the data should be provided. For example, what is the definition of obesity (e.g., BMI>30) or smoking status (e.g., current, ever or non-smoker).

Statistical Analyses:

7. Statistical testing methods for Table 1 should be described.

8. Figure 1 is bar chart not histogram. For age groups, use thresholds e.g., 40, and 70 might be enough for grouping. For Fig 1 A-D, Y-axis should be rate (%) for fair comparison among these groups. Fig 1 C, the sample size of “Patients Needing ICU Level Care” is 52 or 77? For Fig 1E, please remove “unknown” category.

6. PLOS authors have the option to publish the peer review history of their article (what does this mean?). If published, this will include your full peer review and any attached files.

Reviewer #1: No

Reviewer #2: No

---

## [Author Response · Author response to Decision Letter 0]

16 Jun 2020

We appreciate the valuable feedback from the Editors and Reviewers. Herein, we detail the changes to our manuscript that we have made in response to the helpful comments and suggestions provided. We believe that the manuscript has been improved as a result.

Comments from the Editors

Reply: We thank the Editor for this suggestion and have ensured adherence to the style requirements of PLOS ONE.

2. In your ethics statement in the Methods section and in the online submission form, please provide additional information about the data used in your retrospective study. Specifically, please ensure that you have discussed whether all data were fully anonymized before you accessed them and/or whether the IRB or ethics committee waived the requirement for informed consent. If patients provided informed written consent to have data from their medical records used in research, please include this information.

Reply: This has been clarified in the methods section to reflect that the IRB waived the requirement for informed consent.

Data Collection (Page 7): “The CSMC institutional review board approved all protocols for the current study and waived the requirement for informed consent.”

If there are ethical or legal restrictions on sharing a de-identified data set, please explain them in detail (e.g., data contain potentially sensitive information, data are owned by a third-party organization, etc.) and who has imposed them (e.g., an ethics committee). Please also provide contact information for a data access committee, ethics committee, or other institutional body to which data requests may be sent.

Reply: We appreciate this query and can clarify that even after identifiers are removed, the primary dataset includes individual patient level information on the age, sex, race, ethnicity, and date of admission to one of two named hospitals. Given that we specify admissions occurred after March 8, 2020 and over the course of the subsequent several weeks in the setting of the local COVID-19 pandemic, even removal of date of admission would allow possible individual-level identification of patients who live locally within our communities. Given that our data contain this type of potentially sensitive information, there is an ethical restriction to sharing this information publicly.

4. Please amend the manuscript submission data (via Edit Submission) to include authors Joseph E. Ebinger, MD, MS, Natalie Achamallah, MD, MS, Hongwei Ji, MD, Brian L. Claggett, PhD, Nancy Sun, MPS, Patrick Botting, MSPH, Trevor-Trung Nguyen, BS, Eric Luong, MPH, Elizabeth H. Kim, BA, Eunice Park, BS, Yunxian Liu, MS, PhD, Ryan Rosenberry, PhD, Yuri Matusov, MD, Steven Zhao, MD, Isabel Pedraza, MD, Tanzira Zaman, MD, Michael Thompson, BS, MS, Koen Raedschelders, PhD, Anders H. Berg, MD, PhD, Jonathan D. Grein, MD, Paul W. Noble, MD, Sumeet S. Chugh, MD, C. Noel Bairey Merz, MD, Eduardo Marbán, MD, PhD, Jennifer E. Van Eyk, PhD, Scott D. Solomon, MD, Christine M. Albert, MD, MPH and Peter Chen, MD.

Reply: We thank the Editor for this guidance and have now updated the submission data, as directed. 

Reply: We appreciate this comment. As suggested, we have now added these results to the manuscript supplement, and specifically in Supporting Table 6 and Supporting Table 7.

6. We note that Supplemental Figure 1 in your submission contain map image which may be copyrighted. All PLOS content is published under the Creative Commons Attribution License (CC BY 4.0), which means that the manuscript, images, and Supporting Information files will be freely available online, and any third party is permitted to access, download, copy, distribute, and use these materials in any way, even commercially, with proper attribution. For these reasons, we cannot publish previously copyrighted maps or satellite images created using proprietary data, such as Google software (Google Maps, Street View, and Earth). For more information, see our copyright guidelines: http://journals.plos.org/plosone/s/licenses-and-copyright.

Reply: We appreciate this query and can confirm that the map was generated by our team using ArcGIS software by Esri and is not copy written. We have now added to the manuscript a statement regarding the use of this software to the revised Figure Legend, as shown below:

Supporting Figure 1 Legend (Page 26): “The map shown was generated using ArcGIS software by Esri.”

Comments from Reviewer #1

1. In the Data Collection section, the authors mentioned that they obtained data on length of stay, death, and vital signs/laboratory diagnostics assessed within 1 week prior to presentation. I did not find any analytical results related to these variables (as shown in Table 1, baseline characteristics, or adjustment in the regression models). How were these data utilized in the study?

Reply: We thank the Reviewer for the helpful comments and suggestions provided. We especially appreciate this important query regarding the variables that were captured during initial clinical assessment and hospital admission. We recognize, as have prior publications, that many of these clinical variables (e.g. temperature, respiratory rate, oxygen saturation, white blood cell count, interleukin-6, d-dimer, ferritin, chest x-ray findings, etc) directly reflect the severity of illness upon initial clinical presentation – and this degree of illness severity is often dependent on the time between infection onset and initial presentation to medical attention (i.e. the longer the wait before presenting for medical care, the greater the severity of illness on initial presentation). Therefore, because the presenting clinical measures (e.g. vital signs, laboratory values, imaging diagnostics) can be highly variable, we elected to focus this analysis on the pre-existing clinical characteristics that may predispose to Covid-19 illness severity in a manner that is much less dependent on timing of presentation. In addition to removing mention of these more highly variable measures from the Data Collection section, we have also added clarification regarding the main motivation behind the study design: 

Data Collection (Page 6): “Because presenting clinical measures such as vital signs and laboratory values can be highly variable, based on timing of the original clinical presentation, we elected to focus on pre-existing traits that may predispose to Covid-19 illness severity in a manner less dependent on the timing of patients presenting to medical care.”

2. As this study was conducted on the basis of electronic health records, I assumed that some missing data inevitably existed. I suggest briefly summarizing how much data were missing and how the authors treated the missing data in their analyses.

Reply: We thank the Reviewer for this very important point. We also recognize the limits of data gathered from the electronic health record. For this analysis, given the importance of each prioritized variable including in each model, we completed manual chart reviews to extract and verify information for any data variables that originally appears to be missing or outliers in value. This comprehensive approach allowed us to resolve potential missingness. As helpfully suggested by the Reviewer, we have now added clarification of our approach to the revised manuscript:

Data Collection (Page 6): “We conducted iterative quality control and quality assurance analyses on all information extracted from the EHR; all data variables included in the main analyses were verified for completeness and accuracy through manual chart review, to avoid variable missingness or potential impact of inappropriate outliers in statistical modeling.”

3. What model was chosen for performing the ordinal logistic regression analysis? If the proportional odds model was selected, did the authors check the proportional odds assumption?

Reply: We appreciate this important question from the Review. As suggested, we have conducted analyses to test the proportional odds assumption. Specifically, we used the Brant test with the results displayed in the Table R1.3.a below. We observed two predictor variables that demonstrated significant deviation from the proportional odds assumption: hypertension and Elixhauser score. Thus, we further investigated these deviations by conducting sensitivity analyses wherein each component of the primary outcome (an ordinal variable) was treated as a separate binary outcome in separate logistic regression models. The three separate binary outcomes were defined as: Outcome 1 (‘admitted to floor’) = (outcome >=1) vs (outcome =0); Outcome 2 (‘admitted to ICU’) = (outcome >=2) vs (outcome <=1); and Outcome 3 (‘intubation’) = (outcome =3) vs (outcome <=2). As shown in Table R1.3.b below, sensitivity analyses revealed that hypertension is not significantly associated with any of the binary outcomes in the sub-analyses. These findings indicate that potential deviation from the proportional assumption does not substantially impact interpretable results for hypertension, which was also not associated with the primary ordinal outcome in the main analyses. For the Elixhauser score, which was positively associated with the primary ordinal outcome in the main analyses, we observed that it was also consistently positively associated with all three binary outcomes in the sensitivity analyses. Thus, interpretation of an overall positive association with the primary outcome is not threatened. However, sensitivity analyses did reveal that the Elixhauser score appears to have a stronger association with outcome 1 (admitted vs not admitted) than with outcome 3 (intubation vs no intubation). For this reason, we have now added details to the Results section to clarify this important finding:

Results (Pages 10-11): “We used the Brant method to test the proportional odds assumption for consistency of associations across our ordinal outcome; these analyses revealed no substantial qualitative violations, but did indicate that the Elixhauser score was predominantly associated with the specific outcomes of admission versus non-admission (OR 4.34, P<0.001) and need for intensive care versus no intensive care need (OR 1.55, P=0.008) that with the less frequent outcome of needing intubation versus no need for intubation (OR 1.24, P=0.25)."

Table R1.3.a. Results of the Brant test of the proportional odds assumption.

Variables Chi-square Probability

Age 5.967 0.051

Male sex 4.958 0.084

African American race 3.628 0.163

Hispanic ethnicity 0.923 0.630

Obesity 2.184 0.336

Hypertension 9.030 0.011

Diabetes mellitus 1.857 0.395

Elixhauser comorbidity score 16.537 0.000

Prior myocardial infarction or heart failure 0.024 0.988

Prior COPD or asthma 0.857 0.651

ACE inhibitor use 2.426 0.297

Angiotensin receptor blocker use 0.333 0.847

Table R1.3.b. Results of analyses treating each outcome as a binary outcome.

 Outcome 1:

Admitted to Floor Outcome 2: Admitted to ICU Outcome 3: Intubation

Variables OR P value OR P value OR P value

Age 1.55 <0.001 1.37 0.001 1.59 <0.001

Male sex 1.64 0.054 2.86 0.001 4.07 <0.001

African American race 1.66 0.190 2.16 0.046 3.35 0.004

Hispanic ethnicity 1.16 0.673 1.60 0.236 1.85 0.180

Obesity 1.99 0.059 1.65 0.189 2.44 0.039

Hypertension 1.14 0.690 1.46 0.291 0.66 0.325

Diabetes mellitus 2.81 0.006 1.59 0.194 1.47 0.359

Elixhauser comorbidity score 4.34 <0.001 1.55 0.008 1.24 0.249

Prior myocardial infarction or heart failure 0.69 0.601 0.61 0.297 0.61 0.360

Prior COPD or asthma 0.80 0.539 0.60 0.194 0.48 0.128

ACE inhibitor use 0.42 0.119 0.30 0.037 0.49 0.264

Angiotensin receptor blocker use 0.85 0.718 1.11 0.813 1.17 0.749

4. Was there collinearity between any of the predictor variables (e.g., ethnicity and race, which were treated as two different variables) listed in Table 1? As all the variables listed in Table 1 were included in the multivariable regression models and each OR was calculated and interpreted for each independent variable in this study, the authors should consider checking the assumption of no multi-collinearity, which is one of the assumptions when performing ordinal logistic regression analyses.

Reply: We thank the Reviewer for this very thoughtful suggestion. We have now assessed for multicollinearity by calculating the Variance Inflation Factor (VIF) value for each of the variables, all of which were <5, suggesting no multicollinearity. We have now added report of these findings to the revised manuscript.

Statistical Analyses (Pages 7-8): “We calculated the variance inflation factor (VIF) for each of the predictor variables to confirm absence of any substantial multicollinearity.”

Table R1.4. Results of testing for collinearity between predictor variables.

 VIF

Age 1.68

Male sex 1.04

African American race 1.15

Hispanic ethnicity 1.10

Obesity 1.21

Hypertension 1.83

Diabetes mellitus 1.37

Elixhauser comorbidity score 2.09

Prior myocardial infarction or heart failure 1.77

Prior COPD or asthma 1.12

ACE inhibitor use 1.18

Angiotensin receptor blocker use 1.22

5. In Table 2, why did you calculate only the OR of the African American race? There are other races such as Asian and others (I assume the white race as reference) shown in Table 1.

Reply: We appreciate this query and can confirm that we elected to compare risks for the African American race to risks for all other races combined, for two reasons: (1) recently reported concerns of excess risk for African Americans along with limited understanding of whether or not comorbidities contribute to this risk; and, (2) the sample size for certain individual race groups was too small for certain comparisons and particularly for the less common outcome categories such as intubation. We have now added more clarification regarding the approach to analyses of risk by race:

Statistical Analyses (Page 7): “Race was treated as a binary covariate: African American and non-African American. This approach was selected given the recently reported concerns of excess risk for African Americans[29], along with limited understanding of whether or not comorbidities contribute to this risk, in addition to the sample size for other race groups being too small for certain comparisons.”

Table 2, footnote (Page 11): “The referent is non-African American race.”

6. As the authors claimed a trend toward lower illness severity among the patients chronically treated with angiotensin-converting enzyme inhibitor (ACEI) therapy, with an OR of 0.48 (P = 0.06), I suggest describing how they defined “chronically treated with drugs (ACEI and ARB)” in the Methods section, as these findings are interesting and clinically relevant.

Reply: We appreciate this helpful suggestion and can confirm that chronic treatment with these therapies refers to verification of stable ongoing use of the medication. We carefully adjudicated chronic medication use based on the following two criteria: (1) presence of documented use of the medication in a provider clinic note, and (2) active prescription for standing use of the medication in the medication ordering system. We have now added details regarding our approach to adjudicating medication use in the methods section:

Data Collection (Page 6): “Chronic use of ACE or ARB medications was verified by confirming presence of documented ongoing medication use in an outpatient provider’s clinic note along with presence of an active outpatient prescription for the medication, both dated from prior to Covid-19 testing.”

7. On page 10, first paragraph: “…ACE inhibitor (OR 0.38, 95% CI 0.13-0.17, P=0.09)…,” I believe there was an error in the confidence interval. Please correct it. Please check the accuracy of all the statistical values presented in this manuscript.

Reply: We thank the Reviewer for identifying this typographical error. Indeed, as the Reviewer astutely observed, the correct confidence interval values for this estimate should be 0.13 and 1.17. We have now made these corrections to the revised manuscript (Page 12). In addition, we have carefully rechecked all values in the manuscript against the original statistical programming output from R.

Comments from Reviewer #2

1. Recent studies indicate that SARS-CoV-2 might enter host cells by binding angiotensin-converting enzyme 2 (ACE2). However, the theme of this study is to investigate the risk of severity in relation to pre-existing traits rather ACE2. In this sample, the age and race differences exist among the 4 groups. If ACEI or ARB prescriptions are related to age or race, this might cause the bias. Further, the current sample size of ACEI and ARB is too few to confirm ACE2 hypothesis. I recommend the authors should focus on the theme of this study. Particularly, the total percentage of ACEI (n=31) and ARB (n=41) is 16% which is much lower than hypertension 36%. Why is it?

Reply: We thank the Reviewer for the helpful comments and suggestions provided. We agree that the results are certainly not definitive regarding the potential associations of ACEI or ARB treatment with illness severity. Thus, we completely agree with the Reviewer that our findings should be considered hypothesis generating. As helpfully advised by the Reviewer, we have now substantially shortened the discussion of these findings and have re-emphasized the need for additional studies for further investigation. The Reviewer also raises an important question regarding the prevalence of ACEI and ARB medication therapy in the context of prevalent hypertension in our study sample. We can verify that the majority of patients with hypertension who were taking anti-hypertensive medications that were not from the ACEI or ARB classes were being prescribed medications from alternate classes, the most common being diuretics, beta-blockers, and calcium channel blockers. We have now added clarification on the distribution of anti-hypertensive medication use to the revised manuscript:

Discussion (Pages 16-17): “The use of ACE inhibitor or angiotensin receptor blocker (ARB) medications has been a focus of attention given that these agents may upregulate expression of ACE2, the viral point of entry into cells[46] and alveolar type 2 epithelial cells in particular[47]. Alternatively, these agents may confer benefit, given that SARS-CoV-2 appears to reduce ACE2 activity and lead to potentially unopposed excess renin-angiotensin-aldosterone activation[36, 46, 48]. Although we observed a non-significant trend in association of chronic ACE inhibitor treatment with lower Covid-19 illness severity, we found evidence of neither risk nor benefit with ARBs. Together, our findings are supportive of current recommendations to not discontinue chronic ACE inhibitor or ARB therapy for patients with appropriate indications for these medications.”

Results (Page 8): “Of all patients with pharmacologically treated hypertension, a minority were taking ACE inhibitor or ARB class agents and a majority were taking anti-hypertensive medications from alternate classes.”

2. Because there are different forms of ordinal logistic regression models to take care of ordinal outcome (0= not require admission; 1=required hospital admission without intensive care; 2 = required intensive level care without intubation; and, 3 = required intubation during hospitalization), the authors should clearly describe which model is used and examine whether the assumption (e.g., proportional odds) holds or not.

Reply: We appreciate this very thoughtful and important point, which was also raised by Reviewer 1. As shown in Table R1.3.a above, we used the Brant method to test the proportional odds assumption. Because hypertension and the Elixhauser score demonstrated significant deviation from the proportional odds assumption, we conducted sensitivity analyses wherein each component of the primary outcome (an ordinal variable) was treated as a separate binary outcome in separate logistic regression models: Outcome 1 (‘admitted to floor’) = (outcome >=1) vs (outcome =0); Outcome 2 (‘admitted to ICU’) = (outcome >=2) vs (outcome <=1); and Outcome 3 (‘intubation’) = (outcome =3) vs (outcome <=2). As shown in Table R1.3.b above, hypertension was not significantly associated with any of the binary outcomes in the sub-analyses – indicating that potential deviation from the proportional assumption does not substantially impact interpretable results for hypertension, which was also not associated with the primary outcome in the main analyses. For the Elixhauser score, which was positively associated with the primary ordinal outcome in the main analyses, we observed that it was also consistently positively associated with all three binary outcomes in the sensitivity analyses. Thus, interpretation of an overall positive association with the primary outcome is not threatened. However, sensitivity analyses did reveal that the Elixhauser score appears to have a stronger association with outcome 1 (admitted vs not admitted) than with outcome 3 (intubation vs no intubation). For this reason, we have now added details to the Results section to clarify this important finding:

Results (Pages 10-11): “We used the Brant method to test the proportional odds assumption for consistency of associations across our ordinal outcome; these analyses revealed no substantial qualitative violations, but did indicate that the Elixhauser score was predominantly associated with the specific outcomes of admission versus non-admission (OR 4.34, P<0.001) and need for intensive care versus no intensive care need (OR 1.55, P=0.008) that with the less frequent outcome of needing intubation versus no need for intubation (OR 1.24, P=0.25)."

3. The primary analysis is ordinal logistic regression (the outcome is 0,1,2,3), and the secondary analysis is logistic regression analysis (the focus is for specific outcome, a binary variable). The authors need to explain the meaning of the estimated odds ratio from each analysis. Also, since ECI depends on 31 comorbidities including DM, Hypertension and obesity, do the authors assess the collinearity between the covariates?

Reply: We thank for the Reviewer for these astute comments. As suggested, we have now added a clearer explanation of the interpretation of estimated odds ratios for each of the main analyses:

Results (Page 10): “Each estimated OR value represents the increment in higher (or lower) odds of a patient requiring a next higher level of care, for every unit difference in a continuous variable (e.g. per 10 years of age) or for presence versus absence of a given categorical variable (e.g. male sex). In effect, every 10 years of older age was associated with ~1.5-fold higher odds of requiring a higher level of care, and being male versus female was associated with a ~2-fold higher odds of requiring higher level care.”

We also appreciate the important point regarding the ECI and possible collinearity between covariates. We had carefully considered the constituents of the EIC, which is calculated based on weights with each component comorbidity being assigned a weight ranging from -7 to 11 according to the van Walraven algorithm. Given that both diabetes and hypertension have weights of 0 in the ECI calculation, we elected to include these covariates separately in the main models. We have now added clarification regarding this approach to the revised manuscript:

Statistical Analyses (Pages 7-8): “Because hypertension and diabetes are not calculated as substantial contributors to the Elixhauser comorbidity index, we included each of these traits as separate additional covariates in all multivariable-adjusted analyses. We calculated the variance inflation factor (VIF) for each of the predictor variables to confirm absence of any substantial multicollinearity.”

4. The authors need to provide the details of interaction models. Because of small sample size (n=442, particularly n=52 for patients required intubation), the power to detect 11 interaction terms might be low (in Suppl. Tables 4-5). Please explain it better.

Reply: We appreciate this comment and completely agree with the Reviewer that statistical power to detect potential associations in stratified analyses is often limited when the sample size is modest and the outcome event rate is infrequent. For this reason, we elected to use the ordinal multi-level outcome as the primary outcome, rather than a single infrequent outcome event alone, to maximize statistical power. Although we recognize that addition of an interaction term added to the main model is a relatively statistically stable approach, we agree that subsequent models stratifying results by subgroup are more limited in power and thus should be considered hypothesis generating. As suggested, we have now added more details to the revised manuscript to clarify these important points.

Results (Page 13): “In secondary analyses, we used multiplicative interaction terms to assess for effect modification for associations observed in the main analyses (S4 Table). While considered exploratory or hypothesis generating analyses, we found several interactions of potential interest (Fig 4).”

Discussion (Page 17): “The modest size of this early analysis of our growing clinical cohort may have limited our ability to detect potential additional predictors of Covid-19 illness severity, as well as potential interactions or effect modification relevant to the outcomes; thus, further investigations are needed in larger sized samples.”

5. In Table1: according to the Covid-19 Illness Severity outcome ( i.e., 0, 1, 2, 3), the authors should compare the sample characteristics among the 4 disjoint groups to fit their goal. Smoking status could be added although the data might not be complete. Remove “unknown” category for ethnicity and race.

Reply: We thank the Reviewer for these helpful suggestions. We have now made all of these recommended revisions to Table 1.

6. The details of the data should be provided. For example, what is the definition of obesity (e.g., BMI>30) or smoking status (e.g., current, ever or non-smoker).

Reply: We thank the Reviewer for this helpful suggestion. As advised, we have now added to the revised manuscript details regarding definitions of key covariates. 

Methods (Pages 5-6): “For all patients considered to have Covid-19, based on direct or documented laboratory test result and suggestive signs and/or symptoms, we obtained information from the electronic health record (EHR) and verified data for the following demographic and clinical characteristics: age at the time of diagnosis; sex; race; ethnicity; smoking status defined as current versus prior, never, or unknown; comorbidities, including obesity, as clinically assessed and documented by a provider with ICD-10 coding; and, chronic use of angiotensin converting enzyme (ACE) inhibitor or angiotensin II receptor blocker (ARB) medications. Chronic use of ACE or ARB medications was verified by confirming presence of documented ongoing medication use in an outpatient provider’s clinic note along with presence of an active outpatient prescription for the medication, both dated from prior to Covid-19 testing.”

7. Statistical testing methods for Table 1 should be described.

Reply: We appreciate this suggestion and have now added details regarding the statistical test results displayed in Table 1:

Table 1, footnote (Page 9): “P values are for between-group comparisons using the ANOVA test for continuous variables and the chi-square test for categorical variables.”

8. Figure 1 is bar chart not histogram. For age groups, use thresholds e.g., 40, and 70 might be enough for grouping. For Fig 1 A-D, Y-axis should be rate (%) for fair comparison among these groups. Fig 1 C, the sample size of “Patients Needing ICU Level Care” is 52 or 77? For Fig 1E, please remove “unknown” category.

Reply: We appreciate these helpful suggestions. For the age groups definitions, we evaluated the display using less refined cutpoints and found the result to more sparsely display the same information (as shown in Figure R2.8. below); thus, we have elected to retain the original version and would be happy to include the additional figure in the Supplement if requested by the Editors and Reviewer. For Figure 1, we have adjusted the Y axis labels as suggested. We have also added clarification that the sample size for patients needing ICU level care is N=77. In addition, we have removed the “unknown” category as suggested. Thank you again for these helpful suggestions.

Figure R2.8. Age and Sex Distribution of All Patients with Covid-19, Stratified by Admission Status

---

## [Decision Letter · Decision Letter 1]

6 Jul 2020

Pre-existing traits associated with covid-19 illness severity

PONE-D-20-12746R1

Dear Dr. Cheng,

We’re pleased to inform you that your manuscript has been judged scientifically suitable for publication and will be formally accepted for publication once it meets all outstanding technical requirements.

Kind regards,

Yu Ru Kou, PhD

Academic Editor

PLOS ONE

Additional Editor Comments (optional):

Reviewers' comments:

Reviewer's Responses to Questions

**Comments to the Author**

1. If the authors have adequately addressed your comments raised in a previous round of review and you feel that this manuscript is now acceptable for publication, you may indicate that here to bypass the “Comments to the Author” section, enter your conflict of interest statement in the “Confidential to Editor” section, and submit your "Accept" recommendation.

Reviewer #1: All comments have been addressed

Reviewer #2: All comments have been addressed

2. Is the manuscript technically sound, and do the data support the conclusions?

Reviewer #1: Yes

Reviewer #2: Yes

3. Has the statistical analysis been performed appropriately and rigorously? 

Reviewer #1: Yes

Reviewer #2: Yes

4. Have the authors made all data underlying the findings in their manuscript fully available?

Reviewer #1: Yes

Reviewer #2: No

5. Is the manuscript presented in an intelligible fashion and written in standard English?

Reviewer #1: Yes

Reviewer #2: Yes

6. Review Comments to the Author

Reviewer #1: Thank you for inviting me to review this revised manuscript. The authors have well addressed my previous questions and comments.

Reviewer #2: (No Response)

7. PLOS authors have the option to publish the peer review history of their article (what does this mean?). If published, this will include your full peer review and any attached files.

Reviewer #1: No

Reviewer #2: No

---

## [Editor Report · Acceptance letter]

16 Jul 2020

PONE-D-20-12746R1 

Pre-existing traits associated with covid-19 illness severity 

Dear Dr. Cheng:

I'm pleased to inform you that your manuscript has been deemed suitable for publication in PLOS ONE. Congratulations! Your manuscript is now with our production department. 

Kind regards, 

on behalf of

Dr. Yu Ru Kou 

Academic Editor

PLOS ONE